# The Concurrent Acceptability of a Postnatal Walking Group: A Qualitative Study Using the Theoretical Framework of Acceptability

**DOI:** 10.3390/ijerph17145027

**Published:** 2020-07-13

**Authors:** Neli Pavlova, Megan Teychenne, Ellinor K. Olander

**Affiliations:** 1Health Psychology Section, Psychology Department, Institute of Psychiatry Psychology and Neuroscience, King’s College London, London SE5 9RS, UK; 2Institute for Physical Activity and Nutrition (IPAN), School of Exercise and Nutrition Sciences, Deakin University, Geelong 3220, Australia; megan.teychenne@deakin.edu; 3Centre for Maternal and Child Health Research, School of Health Sciences, University of London, London EC1V 0HB, UK; ellinor.olander.1@city.ac.uk

**Keywords:** motherhood, postpartum, walking, social support, acceptability

## Abstract

Walking groups are often enjoyed by postnatal women, but little is known about what makes them an acceptable activity to this group. This study aimed to investigate the acceptability of a postnatal walking group using the Theoretical Framework of Acceptability (TFA). Seventeen postnatal women took part in a walk-along interview during a walking group session. Semi-structured interviews were based on the TFA and findings were analysed deductively onto the seven TFA components. Overall, the walking group was found to be acceptable. Benefits included meeting other mothers and walking in an accessible and convenient location. Women understood the group aim of supporting new mothers and thought it achieved this aim. Most women reported that attending the group was little effort, although for some the timing did not fit well with their baby’s schedule. Participants stated that meeting other mothers and being physically active aligned with their value system. In conclusion, the acceptability of the walking group was found to be multifaceted, of which enjoyment was one part. These findings can be used when advertising other walking groups or physical activities for the postnatal population.

## 1. Introduction

Physical activity in the postnatal period is associated with numerous health benefits including improved mental and physical health [1]. Whilst women agree that it is important to be physically active postpartum [2], few postnatal women engage in the recommended rates of physical activity [3]. Walking is the most commonly reported form of physical activity among postnatal women due to being easy, free and available to all without the need for childcare [4,5]. Further, walking is a safe activity and a suitable interim behaviour before engaging in more vigorous physical activities [4]. Considering these advantages, postnatal walking groups have been suggested as appropriate to help women increase their physical activity. Walking groups also have the added benefit of providing social support to women at a time when they often want to meet other local mothers [5,6]. Research to date suggests that postnatal walking groups are very well-liked [5], although less is known about what specifically makes these walking groups acceptable to postnatal women.

Previous research on the acceptability of postnatal walking groups is weak. Either studies do not measure acceptability [1,2] or it is operationalised as satisfaction and enjoyment with intervention content/delivery [3,4]. Whilst this information can be helpful to walking group leaders and intervention developers, these factors may not fully explain women’s attendance and ignore the value of asking for their views of walking group acceptability [5]. This is problematic, as the acceptability of an intervention is a key factor in the successful implementation of an intervention [6].

Another limitation of past research is that it has not assessed acceptability using a theoretical framework [3,7]. To improve the understanding of intervention acceptability, the Theoretical Framework of Acceptability (TFA) was published [8], providing a systematic approach to define and assess intervention acceptability. This framework defines acceptability as: “*A multi-faceted construct that reflects the extent to which people delivering or receiving a healthcare intervention consider it to be appropriate, based on anticipated or experienced cognitive and emotional responses to the intervention*” (p.4), and identified seven theoretical constructs as sub-components within the overarching construct of acceptability (see Table 1). Previous research has found that participants provided a more varied assessment of intervention acceptability when asked questions related to TFA, compared to being asked about intervention acceptability in general [9]. Thus, the aim of this study was to utilize the TFA to explore concurrent acceptability of a postnatal walking group.

## 2. Materials and Methods

A qualitative exploratory design was used for assessing the acceptability of a postnatal walking group based on the seven constructs of the TFA [8]. Ethical approval for the project was received from City, University of London (Reference MCH/PR/MSc/17-18/02, date: 10/04/2018).

### 2.1. Postnatal Walking Groups

Participants were recruited from one of two walking groups. Both were organised by a national mental health charity and were free to attend. Each walking group held walking sessions once a week in two parks in Greater London, England. According to the charity, the aim of the groups was to facilitate mothers meeting other new mothers for walking and sharing experiences of motherhood. The target audience for the group was postnatal women, but at times pregnant women also attended. Women may have found out about the walking group in different ways, such as online advertisement, referral from healthcare professionals and word of mouth. The walking group was set up by a charity to support the local community, and not for research purposes.

Group 1 was an established group (established for more than 12 months) with approximately 30 regular participants (defined as those who have attended at least three walking sessions over the last year), whilst group 2 was a newer group (established for less than six months) with about seven regular participants. The groups were quite far apart geographically, preventing women from attending both. Both groups had the same format; firstly, meet at a meeting point, located close to a public park where mothers met with their babies and prams. Second, each walking session lasted approximately 45 min with an average walking distance of 3–3.5 km. Walking leaders chose the route for each session and afterwards women were encouraged to socialise with other group members at a local cafe. Both groups had the same structure and did not differ apart from the location.

### 2.2. Participant Recruitment and Data Collection

After obtaining ethical approval and confirmation from the charity that the walkers could be approached for an interview, a researcher (NP) joined a walking group session at each location to introduce the study. This allowed NP to introduce herself and the research study to the potential participants and to get familiar with the structure of the walking sessions. At the following session one week later, the participants were given information about the study and were invited to take part in walk-along interviews. Potential participants were told that the interview would be audio recorded and all findings treated anonymously. The inclusion criteria were: (1) women who have given birth no more than one year before the interview date, (2) aged 16 or over, (3) able to read and speak English fluently. Women were excluded from taking part if they were pregnant or did not fit the inclusion criteria. All participants were recruited in the summer of 2018. NP attended three group 1 sessions and two group 2 sessions.

After written consent was collected, single individual semi-structured walk-along interviews were conducted. Participants were informed that despite it being an individual interview, there could be a possibility that others could overhear the interview. The topic guide was composed of open-ended question items based on the seven components of the TFA [8]. For instance, ‘burden’, was explored with the question of ‘*how easy or difficult is it to participate in the walking group session?*’. Using a walk-along methodology, where the interview takes place whilst both researcher and participant are walking, has been declared a useful strategy for gaining an understanding of people’s context-specific experiences [10] and has been used previously when exploring experiences of walking groups [11]. The interviews can trigger relevant experiences of participating in walking sessions, which might not be otherwise explored in different contexts [10]. All interviews were digitally recorded by a portable audio recorder and transcribed verbatim.

### 2.3. Data Analysis

Framework analysis [12] was used to analyse the findings. This analysis method was chosen to deductively organise data in line with each TFA construct. After familiarisation with the transcripts, data were organised into each construct of the framework, using headings and subheadings. Findings which did not fit in any of the TFA’s constructs included women’s lack of awareness of physical activity recommendations and is not reported in this paper. Initial data analysis was undertaken by the first author (NP), with the last author (EKO), an experienced qualitative researcher, reading through all transcripts and checking the themes extracted. Analysis was done in parallel with data collection, and when saturation was reached, no more participants were invited to take part in the interview. This happened after 17 interviews. All participants have been given a pseudonym to remain anonymous.

## 3. Results

### 3.1. Participants

Eighteen women were approached about participating in the study. One woman declined participation due to personal reasons. Seventeen women (94%) agreed to participate, met the inclusion criteria and provided written informed consent. Fifteen participants attended group 1 and two attended group 2. All women apart from one were first time mothers. Participants were on average 33 years old (range 26 to 42 years), and their baby 6 months old (range 3–48 weeks). Sixteen of the seventeen women identified as White British.

Interview duration ranged from 6 to 24 min, with an average length of 15 min. All of the interviewed women were regular group attendees, except two who were joining for the first time. The first-timers’ views were similar to the women who had attended for longer, and were thus included in the analysis. The views of the women in group 2 were similar to the views of the women in group 1, thus all data was combined.

### 3.2. Qualitative Findings

Overall, the walking group was found to be acceptable to all interviewed participants. Findings are presented below for each TFA construct.

#### 3.2.1. Affective Attitude

This construct is concerned with the women’s feelings about participating in the walking group. All participants reported very positive feelings about attending the walking sessions. Numerous benefits were mentioned, including the opportunity to meet and socialise with other mothers and to feel supported by others. The group was also found to be very welcoming, Claudia (mother of 11 month old baby) gave this example: ‘*And like today, I forgot my changing bag, and a mum said “Do not worry, we have got nappies for you”, you know, it’s very welcoming. I think the culture of the group is very much that everybody watches out for each other*.’ Danielle added that the other mothers at the walking group made her feel understood and ‘*without being judged as a stupid mum’* (Danielle, mother of 6 month old baby). Another participant, Anna (mother of 11 month old baby), also enjoyed attending the group and suggested it should be available in more areas and *‘should be available to everybody who has a baby’*. Whilst technically not part of the walking group, the WhatsApp group a few mothers had set up after meeting at the walking group was also seen as supportive.

Further, the group was found to be convenient to join and its flexibility was well liked. For example, if women were running late they could join the group along the route, as Karis told us, *‘you can always join later which is a key with these little guys’* (Karis, mother of 7 month old baby). Specific to the walking, views differed on the appropriate pace and activities. Petra told us it was good as she ‘*would not go any faster’* (mother of 6 week old baby), while Ines suggested ‘*it would be quite nice to have activities with harder intensity’* (mother of 2 month old baby).

Finally, the walking session was seen as beneficial to the babies. In the words of Barbara, ‘*women could walk and chat with other women, the baby can sleep’,* (Barbara, mother of 11 month old baby). Interestingly, the benefits to the babies were not discussed as much as the benefits to the women themselves, suggesting that this is an activity women primarily thought was for them and not their child.

#### 3.2.2. Self-Efficacy

Findings related to self-efficacy, defined as participants’ confidence to participate in the walking sessions, was divided into attending and completing the walking session. Overall, all participants reported feeling very confident about attending and completing the walking session.

Women were consistent in their reports of possible circumstances preventing them from attending a session. Barriers to attend included the baby being sick or having to attend a medical appointment. Anna reported that ‘*A couple of weeks ago she [daughter] was quite poorly, so we were not able to come then.’* (Anna, mother of 11 month old baby). Other barriers mentioned by the postnatal women were ‘*tiredness and the lack of sleep’,* (Claudia, mother of 11 month old baby) and the timing clashing with the baby’s nap time. For several women, poor weather was a reason for not attending the walking group. For others, poor weather meant they joined the session at the end, when the group went to a café.

All participants reported feeling able to complete the sessions. A hungry or upset baby was reported as a reason to temporarily stop. In the words of Lilian, if her baby was hungry, she found a ‘…*bench and feed, and then catch up with the other ladies’* (Lilian, mother of 1 month old baby). Claudia told us support from the other women was available if one of the babies cried, ‘*I had that, but some people were able to help. Or to hang back and say “Are you all right? Do you want me to hold him while you do this?’* (Claudia, mother of 11 month old baby).

Finally, women reported that the flexibility of ‘*not having to take part in the whole session’,* (Danielle, mother of 6 month old baby) helped them attend. Women appreciated the opportunity *‘to come when you can’,* (Barbara, mother of 11 month old baby).

#### 3.2.3. Intervention Coherence

Intervention coherence is understood as the extent to which participants recognise the aim of the walking group. According to the mental health charity organising the walking group, its aim was to facilitate new mothers meeting other new mothers for walking and sharing experiences of motherhood. This aim was echoed by the majority of the interviewed women, for example, Fran (mother of 6 month old baby) said ‘*I think, the main aim for these sessions are that mums, like, get together, have a chat and, you know.’*

A large number of participants explicitly stated that the group’s aim was to improve the mental health of new mothers ‘I think that the main aim is… is to help women… maintain their mental health. After pregnancy. I do have a baby. I think it would be very easy to slip into depression when your whole life changes and really… It’s a huge, huge change’ (Danielle, mother of 6 month old baby). This perceived aim was based on the name of the group and charity and related to how participants heard about the group. Several participants also recognised that mental health and keeping active were linked and Claudia (mother of 11 month old baby) explained: ‘*I think it’s for mental health. Because of Mindful Mums. And I think walking and mental health are linked. It is important to maintain your sanity when you have a small child and you don’t have the time for yourself…*’. Finally, some participants reported that the aim of the walking groups was to help women be active.

#### 3.2.4. Perceived Effectiveness

This construct is understood as the extent to which the walking is perceived as likely to achieve its purpose. Closely related to women’s views on intervention coherence, participants believed the group helped them meet other mothers, increase their physical activity and improve their mood. Maria (mother of 3 week old baby) told us: *‘I think it [aim of group] is to promote positive well-being and meet new parents./…/you’re always happier afterwards, um yeah. So they definitely do the job they’re supposed to.’*

#### 3.2.5. Burden

This construct is closely related to self-efficacy and focuses on the perceived amount of effort that is required to participate in the intervention. Overall, attendance was perceived to be little effort and ‘*really easy’,* (Fran, mother of 6 month old baby). Many participants reported that the location was convenient, with Nina (mother of 8 month old baby) explaining that *‘…the location is great- we live just 10 minutes’ walk from here’.*

In terms of physical effort, a couple of participants reported that the walking intensity was appropriate. For example, Harriett (7 month old baby] reported: *‘… it’s [the walking pace] perfect. It’s nice handled. If, there’s such a good group, you can be walking quite at the front or you could be handling along at the back. ‘Cause sometimes there are a lot heavy pregnant people here, so if you want a slow walk, you can’*. Lilian (mother of 1 month old baby) told us that ‘*It is quite an easy walk around the park*.’

#### 3.2.6. Opportunity Costs

This concept is understood as the extent to which benefits, profits or values must be given up to engage in the intervention. Overall, the participants overwhelmingly reported that few activities had to be given up to engage in the walking group. For example, women said they could attend, and afterwards: ‘*[I] can still catch up with the cooking and washing after the sessions’* (Maria, mother of 3 week old baby).

For some women, it was easy to attend the walking group as there were no other activities at the same time. In the words of one participant, ‘*during this time of the sessions, there is nothing else’* (Barbara, mother of 11 month old baby).

#### 3.2.7. Ethicality

This construct centres on the extent to which the walking group were perceived to be a good fit with the participants’ value system. All participants reported that being physically active was viewed as valuable. For example, Jordan stated that ‘*I want to be fit enough to kind of keep up with her, without having something which holds me back’* (mother of 7 month old baby), associating being fit with her ability to mother her child. When asked this question, a couple of participants also reported appreciating the sessions giving them an opportunity to meet other mothers and *‘chat with other mums about baby things’* (Olivia, 11 month old baby), which was valued by all participants.

## 4. Discussion

This is the first study to our knowledge to assess the acceptability of a postnatal walking group, using the TFA. Women reported enjoying attending the walking group, similar to previous findings [3,13]. The novel findings of this study include how the participants understood the aims of the walking group, the alternatives to attending and to what extent the group fits within the participants’ value system. These factors added to their overall sense of acceptability of the walking group.

Numerous benefits were mentioned regarding the walking group, the primary one being the opportunity for social support. This is in line with previous research with postnatal women showing the importance of social support, sharing experiences and advice in a non-judgmental environment [14]. Other benefits of the walking group reported by participants included convenient location, ease of activity, no cost and flexibility. Flexible services are likely to be imperative to accommodate postnatal women, who often report difficulty leaving the house on time and an unsettled baby or lack of sleep the previous night, which may change plans quickly [14]. A strength of the walking group was that if a baby became unsettled or hungry, women could attend to their baby and then re-join the group. This level of flexibility must be considered when assessing the fidelity of walking groups. It must also be acknowledged that due to exhaustion and baby routines, feeding and nap times, structured community-based physical activity programs are not always suitable for postnatal women [15]. Some women also mentioned that the time of the groups did not fit very well with their baby’s schedule, suggesting that as babies grow and their schedules change, it may become more difficult for women to attend a walking group. It may also be harder for women to attend when their baby is older and may not want to sit in a buggy for an extended time [13]. A barrier mentioned by some women was the weather, which prevented some of them from attending the walking group. This, again, has been noted previously [1,16].

All the above benefits provided women with high confidence that they could attend the walking group and, when there—take part in the walking. This contributed to the overall assessment that the walking group was acceptable. Little opportunity cost also contributed to this assessment—the participants did not report that they had to give something up to attend the walking group. Rather, for some women, their main aim may have been to leave the house, and the walking group fitted that aim.

Whilst the primary aim of the group was not to increase physical activity per se, this was mentioned as the aim by a couple of the participants. Interestingly, participation in walking groups is associated with increased physical activity in the general population [17] but not postnatal women [1,4,18,19]. This may be due to recruiting physically active participants [11] or the control group increasing their physical activity as much as the intervention group [7]. In our study, many participants suggested that the walking group was to help women with their mental health. Previous research shows that a walking group can have a positive impact on women’s mental health [1]. All this considered, the aim of the group was to provide mothers with an opportunity to meet other mothers. Some women correctly identified this, and all participants reported that the group achieved this aim. The participants’ different responses to the question on intervention coherence was an interesting and novel finding and more research is needed to assess if the perceived aim of the service has implications for attendance and dropout rates. For long-term attendance, it is likely that perceived intervention coherence and perceived effectiveness should correspond and could be explored in the future.

Finally, the walking group was seen to be a good fit with the participants’ value system in that the participants valued being physically active. Further, a few participants associated being physically fit with being a good mother as it would help them to keep up with their child. None of the participants suggested that walking was not an appropriate activity for new mothers. Assessing if an intervention fits with the participants values is rarely done, however it is a construct within the TFA (ethicality). More research is needed on how an intervention fits within participants’ value systems, as it is likely to be important when individuals choose activities.

### 4.1. Strengths and Limitations

This study has a number of strengths. Firstly, theory and frameworks are underused in perinatal health research [20], and thus using the TFA contributes to the field as a whole as well as the topic. Using theoretical frameworks helps facilitate systematic enquiry and gains in knowledge. Secondly, the walk-along interviews provided a real-time assessment of acceptability. Conducting the interviews during the walking group enabled contextual cues to be discussed that may have been missed in a more traditional interview setup. This methodology had the added benefit of making it easy for women to take part in the study. That said, having their babies and/or other mothers present may have acted as a distraction for some women, possibly limiting the disclosed information or ability to reflect on their behaviour. Another strength was that we tested the acceptability of an already existing group, not one that was set up by researchers, as is often the case [4,11]. Thus, our findings can be used to set up other walking groups.

A potential limitation of the current study is the short interviews, some being less than 10 min long. Whilst the interview schedule was brief and the questions straightforward, longer interviews may have given women more time to reflect on their participation in the walking group. However, given that some interviews were shorter in duration due to needing to attend to their baby, this is likely a wide-spread problem for researchers. It is also possible that participants were brief due to not having much to say or were concerned about being overheard, although no participant explicitly said so. Further, whilst concurrent acceptability is important to assess, additional views from women who chose to stop attending or not attend at all could have provided additional information on acceptability. For example, what was not reported in the current study, was the previous findings that walking can be associated with fatigue and some women reporting that it can be difficult to control the pram when walking downhill [14]. These findings may have been identified had women not taking part in the walking group been interviewed.

Finally, it needs to be acknowledged that some of the TFA constructs were found to be very closely related, such as self-efficacy and burden, in this study. As such, what was coded as self-efficacy, others may code as burden. This is likely to be due to the combination of the simple intervention and the participant sample. In this case, the participants reported intervention burden to be low and confidence to participate in the intervention was high. This may be different in other interventions, where the intervention includes more complex behaviours. Therefore, it may be that the TFA may be more useful when assessing more complex interventions.

### 4.2. Future Research and Implications for Practice

The current study clearly suggests that walking groups can be acceptable to postnatal women. Future research could be conducted to compare if these views differ depending on length of attendance. Research could also longitudinally explore if views change from before, during and after attending the walking group and if anticipated acceptability influences attendance. As their babies grow older, women may look for other activities and stop attending the walking group. Whether a walking group can act as a springboard to other physical activity behaviour also needs to be explored. Interviews could also be done with women choosing to not attend these types of groups to assess their views of the perceived acceptability of walking groups. Eliciting views from non-attenders is important to develop acceptable walking groups for postnatal women.

For practice, this research suggests that women attend a walking group for a number of reasons, but that meeting other mothers is a key motivator. Our findings suggest that providing a social element trumps the chance to be active, so, when advertised, the social aspect of these groups need to be highlighted. This social support can be strengthened by a WhatsApp group as used by some of the study participants. Previous research has suggested that women may be reluctant to attend groups with women they do not know, so inviting a friend to take part as well may increase attendance [21]. Providing groups that are flexible is also needed, so that women feel they can attend whilst arriving late or can take a break when their baby becomes unsettled.

## 5. Conclusions

To conclude, findings from this study show that a postnatal walking group is acceptable to its attendees. This acceptability is multifaceted and goes further than enjoyment and satisfaction with ease to attend, understanding the aim of the group and valuing this aim also important. These findings can be used when advertising other walking groups or physical activities for the postnatal population.

## Figures and Tables

**Table 1 ijerph-17-05027-t001:** The Theoretical Framework of Acceptability constructs and their definitions [8].

Acceptability Concept	Definition
Affective attitude	How an individual feels about the intervention
Burden	The perceived amount of effort that is required to participate in the intervention
Ethicality	The extent to which the intervention has a good fit with an individual’s value system
Intervention coherence	The extent to which the participant understands the intervention and how it works
Opportunity costs	The extent to which benefits, profits or values must be given up to engage in the intervention
Perceived effectiveness	The extent to which the intervention is perceived as likely to achieve its purpose
Self-efficacy	The participant’s confidence that they can perform the behaviour(s) required to participate in the intervention

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
