# Peer review of "The Concurrent Acceptability of a Postnatal Walking Group: A Qualitative Study Using the Theoretical Framework of Acceptability"

_ijerph, 2020, doi:10.3390/ijerph17145027_

Round 1

Reviewer 1 Report

The aim of this paper was to utilize theoretical framework of acceptability to explore concurrent acceptability of a postnatal walking group. The results and conclusion were reasonable, the acceptability of the walking group was found to be multifaceted of which enjoyment was one part. But, it should be careful to understand the potential bias in the results of this study. It is admitted that the authors carefully chose the participants and collected the data. To bring out their true intentions, a researcher joined a walking session and conducted walk-along interviews. But, the attendees of this type of intervention seemed to have already possessed the positive feeling of the attending. On the other hand, the participants highly rated to meet other mothers and to feel supported by others rather than physical activities itself. Therefore, for this purpose, intervention of the walking or physical activities seems to be not necessarily. To compare the social communication group without walking intervention could be better to understand the results. Otherwise, I know it is very difficult, to interview the participants who drop-out the walking intervention would draw out the negative feeling of the intervention.

The author selected 2 groups as for established group and a newer group, but the number of the subjects were imbalance, 15 from group 1 and 2 from group 2, and no group comparison were performed. It must be reconsidered the selection of the participants.

Were the interviews conducted with man-to-man or put some participants together? It shall be written clearly, since the answer of the others could be somewhat influenced to the answer. Were the interviews conducted once for each person? If yes, was there any confidence that the feeling would have never changed over a period of months?

The components of the TFA of acceptability concept were somewhat similar; therefore, the results of each components were closely related to each other. The reviewer wondering if the results of these could clearly discriminate into each component.

The flexibility seems to be the key to this kind of intervention. Self-paced walking, free attending the session, no other activities, and no special preparation for tools or money (except for walking shoes), would be the reasons for the attendees could keep up this type of intervention. Especially, that no obligation for taking part in the whole session would be helpful for the postnatal women, the author already mentioned. But, again, it does not necessarily regard to the walking intervention, the other social communication could be fit to these conditions.

Reviewer 2 Report

The manuscript "The concurrent acceptability of a postnatal walking group: a qualitative study using the Theoretical Framework of Acceptability" is a well-written article, with minor suggestions to the authors.

At first, I thought the focus of the study would be on physical activity, but the aim of this kind of intervention (walking groups) is mostly psychological improvements, and it's ok.

I must confess that I was not very enthusiastic about the results as I could not imagine a different result - if women are participating in these groups, of course, they enjoy it, so I could not understand exactly if the authors had a different hypothesis, or if the literature points a different outcome.

Later I understood that, maybe, the intention was actually to test the TFA as an instrument.

I consider that the text is fine as it is but, authors could try to explain a little more how they classified the outcomes. For example:

Self-efficacy, page 4, line 140 - "self-efficacy was high" what does it mean exactly and how to classify as "high"?

The same goes for the second line of the discussion - "high satisfaction", how did the authors get to that conclusion/classification.

Reviewer 3 Report

Comments to the authors

The manuscript reports a qualitative study that examine the acceptability of a postnatal walking group thought an established theoretical framework. Some aspects about methodology must be addressed.

Methods:

- Page 2, line 59: “Ethical approval for the project was received from the authors’ university (Reference MCH/PR/MSc/17-18/02)”. Please, say the name of the University instead of “from the author’s University”.

- How was the information about the walking groups given to the mothers? How were they recruited?

- There was any exclusion criteria for participation?

- There was any difference in the intensity of exercise developed between the walking group 1 and group 2? Please, add a more detailed description of what it meant to participate in one or the other group.

- What was the frequency of the walking sessions?

- What frequency was considered to be classified as a regular participant? Where low adherence participants excluded?

- Where the women informed that the interview was going to be recorded?

- Where the women informed that they were going to be treated anonymously?

- Did the walk-along interviews take place while other participants where listening?

- Where most of the women attending with their babies?

Results:

- Among the 37 participants from both groups, only 18 where approached. Did the other 19 participants meet the inclusion criteria? Why only these 18 where approached?

- Some of the interviews last only 6 minutes. Were all participants asked about all the framework questions?

- Even if this is a qualitative study, could you add information about the percentage of women who answered the most common findings?
